# *CAFÉ*: Coverage-Aware Self-Distillation to Mitigate Forgetting in Deep Networks

## Abstract

Deep neural networks rarely exhibit global overfitting in the classical sense, yet they often suffer from a less visible problem - *forgetting* of previously learned patterns. This phenomenon, which was termed *local overfitting*, degrades performance in specific regions of the input space even as overall accuracy improves. To address this problem, we propose *CAFÉ* (*C*overage-*A*ware *F*orgetting *E*limination) - an *online*, *validation-aware*, *single model* method, which mitigates forgetting during training while exploiting self-distillation. *CAFÉ* identifies and prioritizes checkpoints that uniquely recover forgotten validation samples, dynamically weighting their contributions to form evolving soft labels for each epoch of training. Our experiments show that *CAFÉ* consistently outperforms both standard training and recent self-distillation SOTA methods under clean and noisy labels, across CIFAR-100 and TinyImageNet, with and without data augmentation. Beyond raw accuracy gains, our results provide quantitative evidence of the substantial impact of forgetting on deep learning performance, and demonstrate that targeted mitigation yields measurable robustness.

## 1 Introduction

Overfitting in deep learning is traditionally identified by a rise in validation error after prolonged training. However, recent studies have shown that deep networks can continue to improve the overall validation accuracy while *forgetting* specific subsets of data they had previously classified correctly (Stern et al., 2025). This phenomenon, termed *local overfitting*, is manifested as a loss of correct predictions in certain regions of the feature space, masked by concurrent accuracy gains elsewhere. In other words, while validation accuracy keeps rising, some sub-populations are "forgotten". This effect is especially visible in standard image classification benchmarks, where the con-

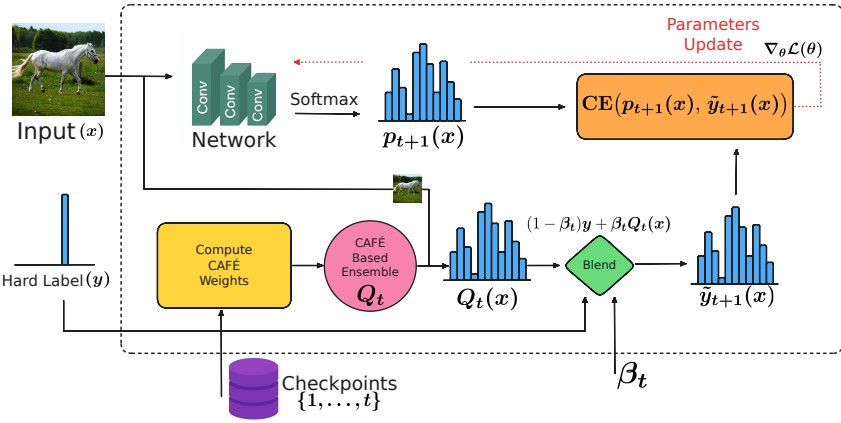

Figure 1: Overview of the *CAFÉ* self-distillation flow at training epoch $t+1$. The network's predictions from epochs $1..t$ are combined with the one-hot label to form a coverage-weighted teacher target for the next update, as described in detail in Section 3.

sequences are particularly severe under noisy labels but can still be observed even in clean datasets. Crucially, this phenomenon is not adequately addressed by early stopping, regularization, or traditional self-distillation.

The problem of *local overfitting* is illustrated in Figure 2, which shows two overlapping bars per checkpoint: one (in blue) shows how many examples are predicted correctly by the current checkpoint's model, and the other (in orange) shows how many of those are predicted correctly by the end of training. Formally, let $\mathcal{V}$ denote the validation set, and define $C_e = \{i \in \mathcal{V} : \text{correct at epoch } e\}$ and $C_E = \{i \in \mathcal{V} : \text{correct at the final epoch } E\}$ The blue bar at epoch $e$ reports the checkpoint accuracy $\frac{|C_e|}{|\mathcal{V}|}$, while the orange bar reports the residual accuracy $\frac{|C_e \cap C_E|}{|\mathcal{V}|}$ that remains valid at the end of training. Their difference,

$$\frac{|C_e|}{|\mathcal{V}|} - \frac{|C_e \cap C_E|}{|\mathcal{V}|} = \frac{|C_e \setminus C_E|}{|\mathcal{V}|},$$

is the *forget gap* - accuracy loss due to examples that the model classified correctly at $e$ but no longer classifies correctly at epoch $E$.

Note that larger gaps indicate more significant forgetting. Note also the difference between Figure 2a - showing the learning curve while training a model with the cross-entropy loss, and Figure 2b - showing the learning curve of the same model trained by our method, *CAFÉ*.

*CAFÉ* reduces the *forget gap* to near-negligible levels. It also removes the characteristic *double-descent* pattern (Nakkiran et al., 2021), seen in Figure 2a as a brief drop in accuracy before recovering, which is absent in Figure 2b.

Traditionally, to mitigate information loss during training, researchers have proposed variants of *temporal ensembling*. These methods update the current loss using an average of past losses or predictions, thereby smoothing the training signal, improving robustness to label noise, and enhancing generalization (see review in Section 2). As an alternative, several approaches construct ensembles of intermediate checkpoints to achieve a similar effect. Building on the emerging notion of *local overfitting*, the specialized *Knowledge Fusion (KF)* method addresses forgetting by selecting a subset of the training checkpoints based on a validation-driven forgetting score and combining them into a weighted ensemble used directly for inference. While effective, this approach faces several notable practical limitations, as discussed below.

In Section 3 we propose a different online approach, which follows the paradigm of *temporal ensembling* while taking into account the specific manifestations of *local overfitting*. Specifically, in *CAFÉ* we adopt a *coverage-based* notion to *steer training online*, introducing a coverage score that weights checkpoints by the *marginal validation coverage* they contribute and forms soft targets accordingly. Although this training-time score differs from the *forget gap*, Figure 2 illustrates the

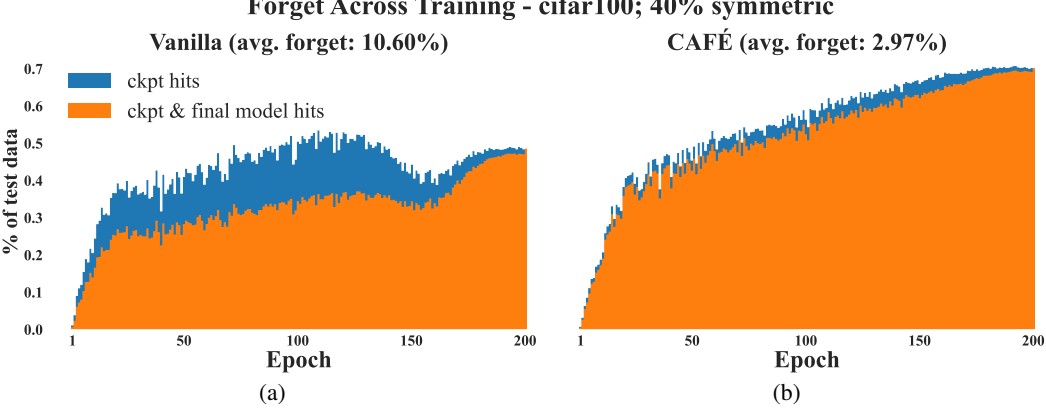

(a)  (b)

Figure 2: Local overfitting. Blue bars denote each checkpoint's validation accuracy, while orange bars indicate the subset of those predictions that remain correct at the end of training. The vertical gap between the bars shows the number of predictions that were temporarily correct but later lost. (a) CE baseline; (b) *CAFÉ*.

practical impact: with *CAFÉ*'s procedure, accuracy is elevated across epochs (and at the end), the double-descent dynamic vanishes, and the forgotten set is markedly reduced. Figure 1 complements this view with a high-level schematic of the *CAFÉ* framework, highlighting the scope within which coverage operates during each epoch, without yet detailing how the weights are computed. In Section 4 we compare our method with a long list of alternative baselines, showing its overall superiority in providing better accuracy while maintaining equal or lower complexity.

*CAFÉ* bridges two previously independent ideas to combat forgetting. On one hand, self-distillation has proven effective for stabilizing training by reusing a model's own predictions; on the other hand, validation performance can reveal which patterns are at risk of being forgotten. By explicitly combining these notions, using validation-driven coverage to decide which past knowledge to preserve, *CAFÉ* transforms self-distillation into a targeted, online mechanism for rolling critical information forward during training. This integration turns insights about local overfitting into a practical, single-model procedure that preserves knowledge without sacrificing efficiency.

Our main contributions are:

- Introduce a *validation-aware coverage score* to weight past checkpoints during training.
- Propose *CAFÉ* (**C**overage-**A**ware **F**orgetting **E**limination), an online self-distillation method using these coverage weights to reduce forgetting.
- Demonstrate superior performance and robustness over strong self-distillation baselines with single-model inference.
- Analyze complexity and provide a variant of *CAFÉ* that mitigates it with minimal checkpoints.

## 2 RELATED WORK

**Leveraging past training information** Several methods smooth the training process by leveraging information from previous epochs instead of relying only on current losses. Temporal ensembling (Laine & Aila, 2017) averages predictions across epochs to improve semi-supervised learning. In noisy-label settings, approaches such as MentorNet (Jiang et al., 2018) and Meta-Weight-Net (Shu et al., 2019) down-weight unreliable samples using historical loss statistics. More recently, EMA-based methods have been shown to improve robustness to noisy labels and generalization (Morales-Brotons et al., 2024). Additionally, checkpoint ensembles have been proposed to aggregate multiple training snapshots, enhancing model performance (Toosifar et al., 2025).

**Forgetting and local overfitting** Networks can lose acquired knowledge late in training, motivating validation-based analyses and remedies (Stern et al., 2025). Knowledge Fusion (KF) addresses this by selecting intermediate checkpoints with a validation-driven forgetting score and aggregating them into a weighted ensemble. While effective, KF is limited by its post-hoc nature, the need to store and evaluate many checkpoints, reliance on exhaustive hyperparameter sweeps to tune ensemble construction, and the natural overhead of maintaining and utilizing an ensemble of models for inference, which together restrict its scalability.

**Self-distillation baselines and scope** We benchmark *CAFÉ* against three main families of self-distillation techniques that rely on *soft targets* but differ in how the "teacher" is formed. (1) **Label-smoothing / teacher-free methods:** Teacher-Free KD (TF-KD) (Yuan et al., 2020) and Zipf-scheduled label smoothing (Zipf's LS) (Zhang & colleagues, 2022). (2) **Current-epoch teachers:** methods that form teachers from intermediate predictions within the same epoch, including BYOT (Zhang et al., 2019), DLB (Shen et al., 2024), CS-KD (Yun et al., 2020), DDGSD (Xu & Liu, 2019), DKS (Sun et al., 2019), FRSKD (Ji et al., 2021), and Self-Distillation with Dropout (Zhu et al., 2022). (3) **Past-epoch teachers:** methods that aggregate predictions across training time, such as PS-KD (Kim et al., 2020), SAT (Huang et al., 2020), EWR-KD (Xia & Yang, 2021), TSD (Liu et al., 2024), and KF (Stern et al., 2025). For completeness we also report Born-Again Networks (BAN) (Furlanello et al., 2018), a widely used multi-round self-distillation baseline.

**How the baseline methods construct soft targets** *(1) Label-smoothing / teacher-free.* TF-KD reframes distillation as label smoothing without a teacher, while Zipf's LS shapes non-target probabilities via a Zipf distribution at negligible cost. *(2) Current-epoch teachers.* BYOT transfers from

deeper to shallower heads; DLB regularizes with last mini-batch predictions; CS-KD enforces class consistency; DDGSD aligns augmented views; DKS adds auxiliary branches with pairwise consistency; FRSKD refines features/logits through an auxiliary branch; and SD with Dropout ensembles stochastic subnetworks. *(3) Past-epoch teachers.* PS-KD aggregates prior predictions; SAT uses momentum-averaged targets with reweighting; EWR-KD adds uncertainty-aware reweighting; TSD builds tolerant targets from stored predictions; and KF ensembles checkpoints via validation-driven forgetting scores. BAN retrains successive student generations.

In contrast, *CAFÉ* is guided by *validation coverage of forgotten examples* for online self-distillation within a single training run. Unlike KF, it avoids post-training checkpoint storage, ensemble inference overhead and costly hyperparameter sweeps, while retaining single-model inference.

## 3 METHOD

*CAFÉ* trains a *single* network in a single pass. It begins by asking the following question at each epoch: *Which earlier checkpoints captured useful patterns that we might have forgotten?* To answer this question, we developed a notion of coverage with respect to a small held-out validation set, which is used to estimate test-set coverage[1]. If a checkpoint correctly classifies validation examples that subsequent checkpoints err on, we say that it provides *marginal coverage*. This *coverage* score determines the influence assigned to the respective checkpoint, when forming the soft targets that guide subsequent training.

A lightweight variant that significantly reduces space and time complexity is described in Section 3.3. The performance difference between the two variants is examined in the ablation study (Table 7). Section 5 summarizes the theoretical justification and complexity analysis, while Appendix D contains the complete derivations and proofs.

### 3.1 BASIC *CAFÉ* METHOD

Algorithm 1 presents the procedure used by *CAFÉ* to generate blended targets after completing epoch $t$. These targets are subsequently used for optimization in epoch $t + 1$. Each step of this per-epoch computation is described in detail in the remainder of this section.

---

**Algorithm 1** *CAFÉ*: epoch $t$

---

**Input**: Validation set $\mathcal{V}$; coverage subsets $\{C_s\}_{s=1}^t$; validation accuracies $\{a_s\}_{s=1}^t$; checkpoint predictions $\{p_s(x)\}_{s=1}^t$; scheduler $\beta_t \in [0, 1]$.
**Output**: Blended target for next epoch optimization.
  1: **Order checkpoints:** $S_t \leftarrow \{1, \dots, t\}$ sorted by $a_s$ (high $\rightarrow$ low)
  2: **Marginal coverage sweep:** $U \leftarrow \emptyset$; **for** $s \in S_t$ **do** $\Delta_s \leftarrow |C_s \setminus U|$;   $U \leftarrow U \cup C_s$
  3: **Normalize coverage gains:** $\widehat{\Delta}_s \leftarrow \Delta_s \big/ \sum_{j \in S_t} \Delta_j$
  4: **Construct coverage-weighted teacher:** $Q_t(x) \leftarrow \sum_{s \in S_t} \widehat{\Delta}_s \, p_s(x)$
  5: **Form blended target for next epoch:** $\tilde{y}_{t+1}(x) \leftarrow (1 - \beta_t) \, \text{one-hot}(y) + \beta_t \, Q_t(x)$
  6: **Return** $\tilde{y}_{t+1}(x)$

---

STEP 1: ORDER CHECKPOINTS

All checkpoints in the range $s \in [1 \dots t]$ are sorted based on their accuracy on the validation set $\mathcal{V}$, providing the sorted list $S_t = \{s_j\}_{j=1}^t$. The method then sweeps through this ordered list from beginning (most accurate) to end (least accurate).

STEP 2: MARGINAL COVERAGE SWEEP

While sweeping through list $S_t$, we assign to each checkpoint its marginal coverage of the validation set. More specifically, the first checkpoint $s_1$ (with the highest accuracy) is assigned $|C_{s_1}|$ - the

---

[1]Appendix E provides empirical evidence showing that validation accuracy closely tracks test accuracy even under label noise, justifying its use as a surrogate for generalization.

number of points that it correctly classifies in the validation set. The second checkpoint $s_2$ (with the second highest accuracy) is assigned $|C_{s_2} \setminus C_{s_1}|$ - the number of points in the validation set that it correctly classifies, and that checkpoint $s_1$ does **not** classify correctly. This continues until all the checkpoints are assigned a non-negative value, according to the following formulae:

$$\Delta_{s_i} = \big| C_{s_i} \setminus \bigcup_{j=1}^{i-1} C_{s_j} \big|. \tag{1}$$

Figure 3 illustrates how marginal coverage is computed after sorting checkpoints by validation accuracy. We relabel the sorted checkpoints as $s_1$, $s_2$, and $s_3$, so that $|C_{s_1}| \geq |C_{s_2}| \geq |C_{s_3}|$. The left-hand panel shows the validation examples each checkpoint correctly classifies. Since $C_{s_1}$ and $C_{s_2}$ do not overlap ($C_{s_1} \cap C_{s_2} = \emptyset$), their marginal coverages equal their total sizes: $\Delta_{s_1} = |C_{s_1}|$ and $\Delta_{s_2} = |C_{s_2}|$. Checkpoint $s_3$ overlaps with earlier ones, so only its unique contribution $C_{s_3} \setminus (C_{s_1} \cup C_{s_2})$ counts, giving $\Delta_{s_3} < |C_{s_3}|$. This highlights the dependence on ordering, ensuring that weaker checkpoints are credited only for patterns not already captured by stronger ones.

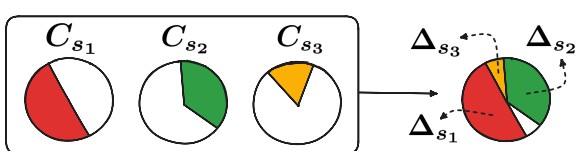

Figure 3: Illustration of marginal coverage: checkpoints are ranked by validation accuracy (left), then their unique contributions are computed (right) to produce $\Delta_s$ values, to be used for weighting.

STEPS 3 AND 4: COVERAGE-WEIGHTED TEACHER

Next, we define the *teacher distribution* $Q_t(x)$ by averaging checkpoint predictions with weights proportional to their *marginal validation coverage*; this distribution is blended later with the hard label in the loss:

$$Q_t(x) = \sum_{s \in S_t} \widehat{\Delta}_s \, p_s(x), \qquad \widehat{\Delta}_s = \frac{\Delta_s}{\sum_{j \in S_t} \Delta_j}, \tag{2}$$

where $S_t$ is the set of checkpoints considered up to epoch $t$, $p_s(x)$ their post-softmax class probabilities (prediction vector), and $\Delta_s$ the marginal coverage counts computed in Step 2.

## 3.2 *CAFÉ* LOSS AT EPOCH $t+1$

We obtain the *CAFÉ* loss by blending the teacher with the hard label using a *parameter-free* exponential schedule:

$$u_t = \frac{t-1}{E-1} \in [0,1], \qquad \beta_t = \frac{1 - \exp(-2\,u_t)}{1 - \exp(-2)}.$$

Here, $E$ denotes the total number of training epochs, so that $u_t$ linearly interpolates between 0 and 1 over the course of training.

Gradually increasing reliance on the soft teacher has been shown to improve stability and generalization in self-distillation, as demonstrated in PS-KD (Kim et al., 2020), where early training leans on hard labels for strong corrective signals and later training benefits from the structural information in soft targets. We therefore choose an exponential rise for $\beta_t$, mirroring the typical exponential decay of train error (Hestness et al., 2017), so that reliance on $Q_t(x)$ grows smoothly as the model stabilizes. The constant "2" is there to ensure a fast-then-plateau curve.

Given $Q_t(x)$ and $\beta_t$, the per-sample target $\tilde{y}_{t+1}(x)$ and loss $\mathcal{L}_{t+1}$ can now be derived:

$$\tilde{y}_{t+1}(x) = (1 - \beta_t)\,\text{one-hot}(y) + \beta_t\,Q_t(x), \qquad \mathcal{L}_{t+1} = \text{CE}\big(p_{t+1}(x), \tilde{y}_{t+1}(x)\big),$$

Above CE denotes the cross-entropy, namely, the $KL$ divergence to $\tilde{y}_{t+1}(x)$ up to a constant. This simple convex combination lets the hard label anchor the semantics, while the coverage-weighted teacher injects structure that preserves information otherwise prone to forgetting.

## 3.3 LIGHT *CAFÉ*

At epoch $t$, *CAFÉ* computes the *marginal validation coverage* $\Delta_s$ of all past checkpoints and forms a coverage-weighted teacher as in (2). We introduce a lighter, more efficient variant - **Light *CAFÉ*** - that uses a tolerance parameter $\tau \geq 0$ to prune checkpoints with negligible marginal coverage. When $\tau = 0$, **Light *CAFÉ* is provably equivalent to Basic *CAFÉ*** (see Appendix D.2, Claim 1), yet it dramatically reduces storage and computation requirements. The complete pseudocode can be found in Appendix A, while empirical effects are analyzed in Appendix C.5.

**Difference from Basic *CAFÉ*:**  Consider set $S_t$ in (2), and let $\mathcal{A}_t \subseteq S_t$ denote the set of *contributing checkpoints*, which are all checkpoints for which $\Delta_s > \tau$ for some $\tau \geq 0$. In the lightweight version of *CAFÉ*, teacher $Q_t(x)$ is redefined in Step 4 as follows:

$$Q_t(x) \;=\; \sum_{s \in \mathcal{A}_t} \widehat{\Delta}_s \, p_s(x), \qquad \widehat{\Delta}_s \;=\; \frac{\Delta_s}{\sum_{j \in \mathcal{A}_t} \Delta_j}. \tag{3}$$

Table 7 empirically demonstrates that a conservative choice of $\tau$ can shrink the size of the contributing checkpoints set ($|\mathcal{A}_t|$) by orders of magnitude relative to $|S_t|$, with $\tau = 0.01$ yielding roughly a 90% reduction in storage space without any significant loss of performance.

## 4 EXPERIMENTS

### 4.1 MAIN RESULTS

We evaluate *CAFÉ*[2] on CIFAR-100 (Table 1) and TinyImageNet (Table 2) using ResNet-18, under clean labels as well as symmetric, asymmetric, and human label noise (CIFAR-100N). Comparisons include standard cross-entropy (CE) with early stopping, SAT, and the KF ensemble. We specifically include SAT because, among self-distillation methods in the literature, it is the strongest-performing approach we could identify that explicitly targets noisy-label scenarios. As DLB does not report matching numerical results, we defer comparison to Section 4.3. Across all settings, *CAFÉ* consistently surpasses these baselines while retaining single-model cost.

Table 1: **CIFAR-100N and CIFAR100.** Mean test accuracy (%, error over 3 random seeds) on image classification datasets using a ResNet-18 backbone, comparing our method and baselines. The best performer in each column is highlighted in bold. The last row shows the improvement of the best performer over vanilla ERM with early stopping. We note that KF results have been copied verbatim from the original paper, so unreported experiments are shown as missing.

| Method / Dataset | CIFAR-100N | CIFAR100 (Symmetric Noise) | | | CIFAR100 (Asymmetric Noise) | |
| --- | --- | --- | --- | --- | --- | --- |
| noise level | 40% human | Clean | 20% | 40% | 60% | 20% | 40% |
| *Vanilla + Early Stopping* | $54.53 \pm .44$ | $78.66 \pm .11$ | $64.98 \pm .11$ | $59.37 \pm .36$ | $50.92 \pm .48$ | $66.61 \pm .12$ | $49.57 \pm .10$ |
| *SAT* | $53.18 \pm .43$ | $77.84 \pm .01$ | $71.97 \pm .10$ | $66.67 \pm .09$ | $55.79 \pm .91$ | $74.43 \pm .07$ | $63.26 \pm .13$ |
| *KF Ensemble* | — | $79.13 \pm .14$ | $72.80 \pm .10$ | $67.00 \pm .10$ | — | $74.20 \pm .10$ | $62.10 \pm .50$ |
| CAFÉ | $\mathbf{63.92} \pm .08$ | $\mathbf{80.25} \pm .05$ | $\mathbf{74.09} \pm .01$ | $\mathbf{69.75} \pm .14$ | $\mathbf{59.42} \pm .26$ | $\mathbf{75.61} \pm .18$ | $\mathbf{67.17} \pm .10$ |
| *Improvement (over vanilla)* | $\mathbf{9.39} \pm .45$ | $\mathbf{1.59} \pm .12$ | $\mathbf{9.11} \pm .11$ | $\mathbf{10.38} \pm .39$ | $\mathbf{8.5} \pm .54$ | $\mathbf{9.0} \pm .22$ | $\mathbf{17.6} \pm .14$ |

Table 2: **TinyImageNet.** Mean test accuracy, see caption of Table 1.

| Method / Dataset | TinyImageNet (Symmetric Noise) | | | TinyImageNet (Asymmetric Noise) | |
| --- | --- | --- | --- | --- | --- |
| noise level | Clean | 20% | 40% | 60% | 20% | 40% |
| *Vanilla + Early Stopping* | $65.37 \pm .01$ | $56.32 \pm .22$ | $49.62 \pm .23$ | $40.02 \pm .38$ | $58.12 \pm .18$ | $43.42 \pm .17$ |
| *SAT* | $63.10 \pm .20$ | $60.59 \pm .17$ | $54.01 \pm .20$ | $39.92 \pm .23$ | $62.45 \pm .02$ | $52.73 \pm .10$ |
| *KF Ensemble* | $68.50 \pm .36$ | $62.80 \pm .20$ | $57.00 \pm .50$ | — | — | — |
| CAFÉ | $\mathbf{68.91} \pm .01$ | $\mathbf{63.20} \pm .03$ | $\mathbf{58.20} \pm .16$ | $\mathbf{47.12} \pm .20$ | $\mathbf{64.71} \pm .03$ | $\mathbf{53.69} \pm .04$ |
| *Improvement (over vanilla)* | $\mathbf{3.54} \pm .01$ | $\mathbf{6.88} \pm .22$ | $\mathbf{8.58} \pm .28$ | $\mathbf{7.10} \pm .43$ | $\mathbf{6.59} \pm .18$ | $\mathbf{10.27} \pm .17$ |

**Augmentation compatibility.**  *CAFÉ* retains its accuracy gains under modern augmentations such as CutMix; see Appendix B for details.

---

[2]Unless stated otherwise, all headline results in this paper use the Basic *CAFÉ* without thresholding

## 4.2 COMPREHENSIVE METHOD-WISE COMPARISON ON CIFAR-100

Table 3 aggregates state-of-the-art CIFAR-100 (clean) results into a single comparison with *CAFÉ*. It reports *Top-1 error*, *Top-5 error*, and *AURC* (confidence quality; (Geifman et al., 2018)) for three commonly used backbones: ResNet-18, ResNet-101, and DenseNet-121. Competitor numbers are taken verbatim from prior work, PS-KD (Kim et al., 2020); TSD and related self-distillation baselines (Liu et al., 2024); label-smoothing variants and DGD (Zhang & colleagues, 2022), with no re-training or re-tuning on our side.

Across all three metrics, *CAFÉ* exhibits a consistent and marked advantage over prior methods.

Although PS-KD achieves competitive performance, it differs from *CAFÉ* in that it uniformly aggregates past predictions, while *CAFÉ* constructs coverage-weighted teachers that emphasize forgotten examples. This targeted weighting produces different soft targets and underlies the accuracy improvements observed in Table 3.

Table 3: **Cross method comparison on CIFAR100.** Rows are ordered by *ResNet-18 Top-1 error* in descending order (worst at top). For all metrics - Top-1, Top-5, and AURC - *lower values indicate better performance*. The Top-1 value for each method is the best (lowest) reported across sources (see text): standard error is included only when the source (Zipf or TSD) provides it, while Top-5/AURC values are taken from PS-KD when available; The superscripts (source) next to method names indicate the paper(s) from which each row was taken. Dashes indicate unreported values. Bold highlights the best (lowest) entry per column within each architecture.

| Method (best across sources) | ResNet-18 Top-1 Err (%) | ResNet-18 Top-5 Err (%) | ResNet-18 AURC ($\times 10^3$) | DenseNet-121 Top-1 Err (%) | DenseNet-121 Top-5 Err (%) | DenseNet-121 AURC ($\times 10^3$) | ResNet-101 Top-1 Err (%) | ResNet-101 Top-5 Err (%) | ResNet-101 AURC ($\times 10^3$) |
|---|---|---|---|---|---|---|---|---|---|
| Vanilla[TSD] | 23.76 ± .07 | 6.90 | 67.65 | 20.05 | 4.99 | 52.21 | 20.75 | 5.28 | 55.45 |
| DGD[Zipf] | 23.52 ± .13 | — | — | 21.82 ± .20 | — | — | — | — | — |
| DDGSD[TSD] | 23.39 ± .47 | — | — | — | — | — | — | — | — |
| BAN[Zipf] | 23.04 ± .04 | — | — | 21.61 ± .14 | — | — | — | — | — |
| SD-Dropout[TSD] | 23.00 ± .00 | — | — | — | — | — | — | — | — |
| TF-KD[Zipf/PS-KD] | 22.71 ± .15 | 6.01 | 61.77 | 19.88 | 5.10 | 69.23 | 20.10 | 5.10 | 58.80 |
| Zipf's LS[TSD] | 22.62 ± .32 | — | — | — | — | — | — | — | — |
| FRSKD[TSD] | 22.29 ± .14 | — | — | — | — | — | — | — | — |
| BYOT[TSD/Zipf] | 22.12 ± .19 | — | — | 21.61 ± .05 | — | — | — | — | — |
| TSD[TSD] | 21.46 ± .35 | — | — | — | — | — | — | — | — |
| DKS[TSD] | 21.36 ± .25 | — | — | — | — | — | — | — | — |
| CS-KD[PS-KD] | 21.30 | 5.70 | 56.56 | 20.47 | 6.21 | 73.37 | 20.76 | 5.62 | 64.44 |
| LS[PS-KD] | 20.94 | 6.02 | 57.74 | 19.80 | 5.46 | 91.06 | 19.84 | 5.07 | 95.76 |
| PS-KD[PS-KD/TSD] | 20.82 ± .23 | 5.10 | 52.10 | 18.73 | **3.90** | 45.55 | 19.43 | 4.30 | 49.01 |
| *CAFÉ*[ours] | **19.75** ± .05 | **4.25** ± .03 | **50.76** ± .02 | **18.67** ± .08 | 3.95 ± .03 | **44.33** ± .88 | **18.04** ± .10 | **3.65** ± .07 | **42.81** ± .34 |

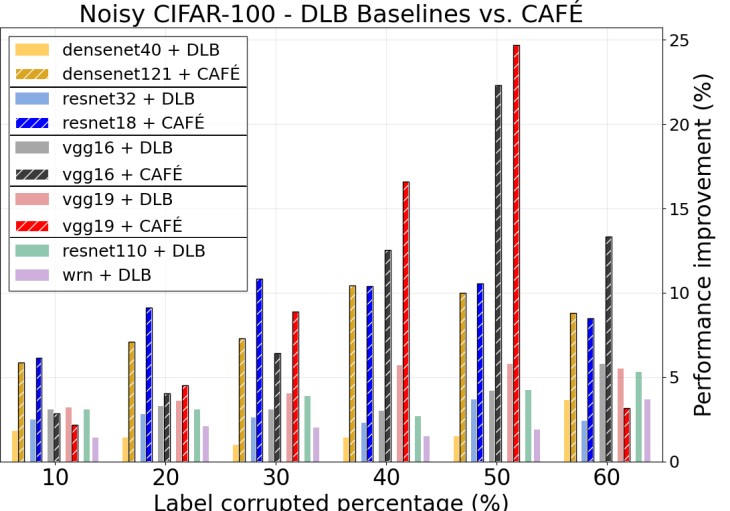

Figure 4: CIFAR-100 with symmetric label noise, relative gains. We compare DLB and *CAFÉ* on matching architecture families for 4 family pairs: ResNet, VGG-16, VGG-19, and DenseNet. In each noise-level cluster, pairs are DLB then *CAFÉ*; slim gaps separate families.

### 4.3 ADDITIONAL COMPETING METHODS

Shen et al. (2024) provide another method (DLB) for targeting the noisy-label regime. However, in this work noisy-label performance is reported through visual plots on custom backbones, without providing tabulated values for standard `torchvision` models. Since *relative gains* are plotted, this format naturally supports reproducibility: we adopt their presentation style and overlay our results using architectures from the same families, ensuring fairness while avoiding the confounding effects of training differences, see Figure 4.

Numerically, averaged over all noise levels, *CAFÉ* delivers relative accuracy gains of up to +18.9 points over DLB, with a mean advantage of about +6.5 points per architecture family and a win rate of approximately 88% across all individual comparisons. These values demonstrate a clear and substantial benefit across the noise scale.

### 4.4 FORGET ANALYSIS

After establishing that our method generalizes well across diverse settings, we next investigate *how* its training dynamics evolve. To begin with, we conclude from Figure 2 that across settings, *CAFÉ* substantially reduces the average forgotten fraction, as the blue–yellow *forget gap* shrinks substantially and the epoch-wise double ascent disappears.

**Checkpoint "life expectancy"**  Figure 5b provides a visualization of how long each checkpoint remains uniquely useful as training proceeds. Read each vertical bar as the *lifespan* of a checkpoint: its height counts how many future epochs still credit that checkpoint with nonzero *marginal validation coverage*. For example, if a bar of height 30 sits above epoch $x=48$, it indicates that the model at epoch 48 retains unique credit until epoch 78; at that point, later models fully inherit its coverage and its marginal contribution falls to zero. Color intensity encodes the magnitude of this marginal coverage. Shading along the bar fades with height, reflecting that a checkpoint's uniqueness typically decays as training proceeds.

When comparing the top panel (vanilla training) and the bottom panel (*CAFÉ*), the contrast is stark: Vanilla training shows many very tall, slow-fading bars, indicating that later checkpoints repeatedly fail to absorb earlier knowledge. In contrast, the bottom panel (*CAFÉ*) shows much shorter, fast-fading bars, indicating rapid handoff of useful patterns. It seems that *CAFÉ* turns "lingering" uniqueness into *transferred* knowledge.

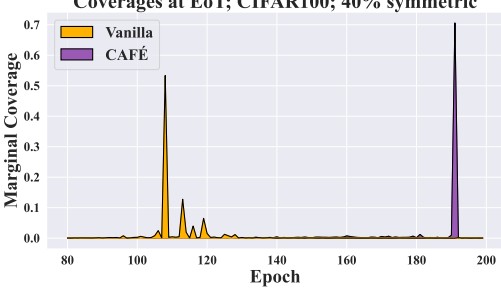

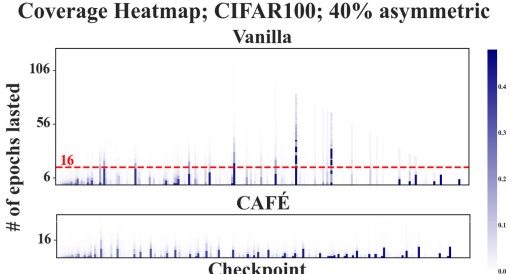

(a) Marginal coverage at the end of training. CIFAR-100, 40% symmetric noise. Yellow: Vanilla; Purple: *CAFÉ*. Epochs prior to 80 are omitted, as their marginal coverage is negligible.

(b) Checkpoint life expectancy. Each bar shows how long a checkpoint remains uniquely useful. Vanilla retains long-lasting uniqueness, while *CAFÉ*'s bars fade quickly, indicating faster transfer of knowledge to later checkpoints.

Figure 5: Two perspectives on forgetting. (a) The epochs that still retain unique coverage at the end of training (EoT). *CAFÉ* shortens these lifespans and concentrates coverage toward later epochs, indicating effective forward knowledge transfer. (b) Vanilla exhibits checkpoints whose unique coverage persists far into training, while *CAFÉ* rapidly transfers this coverage forward, resulting in much shorter checkpoint lifespans.

**End-of-training (EoT) marginal coverage** A complementary view is shown in Figure 5a, which plots, for each epoch, its *marginal coverage measured at the end of training*. The yellow curve (Vanilla) reveals several large spikes originating from mid-training checkpoints: even at the end of training, substantial portions of the validation examples remain *uniquely* attributed to those earlier epochs, indicating that later models never fully inherit their coverage. In contrast, the purple curve (*CAFÉ*) concentrates almost all its mass near the final epochs, indicating that knowledge is systematically rolled forward and embedded into the final model. This aligns with the life-expectancy picture in Figure 5b: *CAFÉ* turns long-lived, lingering uniqueness into a transferred signal.

To address the possibility that the strongest performance occurs mid-training, the quantitative comparisons reported in Section 4.1 evaluate Vanilla under an oracle early-stopping protocol that selects the epoch with the highest validation accuracy, corresponding *precisely* to the largest end of training spike in Figure 5a. Despite this favorable baseline, *CAFÉ*, which explicitly rolls that knowledge forward to later checkpoints, achieves higher test accuracy and substantially lower forgetting, indicating better generalization than freezing at a mid-training checkpoint.

### 4.5 ABLATION SUMMARY

Due to space constraints, the full ablation studies and their empirical evidence have been moved to Appendix C. Below we provide only the headline findings from all ablations:

- **EMA variants.** *Parameter-space* EMA variants provide no extra benefit, coverage in *prediction space* matters most (see C.1).
- **Coverage vs. raw accuracy.** Simply relying on *raw validation accuracy* to guide training is insufficient; the key contributor to performance is constructing teacher weights based on *marginal coverage* (see C.2).
- **Checkpoint order: accuracy vs. chronology.** Sweeping through past checkpoints in reverse chronological order (instead of sorting by validation accuracy) yields worse results, underscoring the importance of using validation accuracy to rank historical models (see C.3).
- **Temporal smoothing.** Smoothing across adjacent checkpoints before weighting offers no gain; validation-guided coverage already ensures stable soft labels (see C.4).
- **Light *CAFÉ* variant.** A small threshold maintains accuracy and cuts storage overhead (and the number of checkpoints kept concurrently) by more than an order of magnitude (see C.5).
- **Blending schedule ($\beta$).** Fixed mixtures ($\beta = 0$ or $\beta = 1$) severely under-perform, showing that neither ignoring the teacher nor relying on it exclusively is effective. In contrast, any smooth increasing schedule (linear, cosine, exponential with different $k$) yields nearly identical accuracy, indicating that the precise functional form is not important as long as $\beta_t$ ramps up over training (see C.6).

## 5 THEORETICAL & COMPLEXITY ANALYSIS

**Complexity** The computational cost of Basic *CAFÉ* is $\mathcal{O}(n \log n + nm)$ in time and $\mathcal{O}(n)$ in space, where $n = |S_t|$ is the number of stored checkpoints and $m$ is the validation set size. For Light *CAFÉ*, the cost depends on the size of the contributing set $\mathcal{A}_t \subseteq S_t$, which shrinks substantially for even small values of $\tau$. A full step-by-step complexity derivation is provided in Appendix D.1.

**Theoretical Guarantee** We prove that once a checkpoint's marginal coverage becomes zero at some epoch, it can never regain positive marginal coverage at later epochs. Consequently, Light *CAFÉ* with $\tau = 0$ is provably equivalent to Basic *CAFÉ* while achieving a significant reduction in storage. Larger reduction is obtained when $\tau > 0$. The formal claim and complete proof are given in Appendix D.2.

## 6 SUMMARY AND DISCUSSION

We introduced *CAFÉ*, an online, validation-aware self-distillation method designed to combat forgetting in deep neural networks. By explicitly weighting past knowledge by its unique contribution to recovering forgotten validation samples, *CAFÉ* improves both robustness and generalization.

Looking ahead, it may be even more powerful to first identify the patterns and examples most prone to forgetting and then develop strategies to prevent their loss, rather than solely attempting to recover forgotten knowledge after the fact.

Our findings support two claims: (1) Forgetting is a pervasive and quantitatively large effect in deep learning, even without label noise. (2) Targeted mitigation via validation-aware weighting of historical knowledge yields measurable improvements in robustness and accuracy. In contrast to many self-distillation methods, *CAFÉ* selects *which* past predictions to trust based on their unique validation coverage, and it retains single-model inference efficiency.

## REPRODUCIBILITY STATEMENT

We specify the algorithm and loss in Section 3 (including the light variant). The full experimental setup - datasets, architectures, training schedules, augmentations, metrics, hyperparameters, and hardware - is documented in Section 4 and Appendix F, with the label-noise protocol in Appendix G. Results are reported as mean $\pm$ s.e. over three seeds. Robustness is evaluated through comprehensive ablations and additional metrics, and theoretical assumptions, guarantees, and complexity bounds are summarized in Section 5 with complete proofs in the Appendix.

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

APPENDIX

## A  LIGHT *CAFÉ* PSEUDOCODE

For completeness, we provide the full Light *CAFÉ* procedure introduced in Section 3.3.

---

**Algorithm 2** Light *CAFÉ*: epoch $t$

---

**Input**: Validation set $\mathcal{V}$; contributing set $\mathcal{A}_{t-1}$; coverage subsets $\{C_s\}_{s \in \mathcal{A}_{t-1} \cup \{t\}}$; validation accuracies $\{a_s\}_{s \in \mathcal{A}_{t-1} \cup \{t\}}$; checkpoint predictions $\{p_s(x)\}_{s \in \mathcal{A}_{t-1} \cup \{t\}}$; scheduler value $\beta_t \in [0, 1]$; tolerance parameter $\tau \geq 0$.
**Output**: blended target for next epoch optimization.
 1: **Order checkpoints:** $\mathcal{K} \leftarrow \{\mathcal{A}_{t-1} \cup \{t\}\}$ sorted by $a_s$, $s \in \mathcal{A}_{t-1} \cup \{t\}$ (high $\rightarrow$ low)
 2: **Marginal coverage sweep:** $U \leftarrow \emptyset$; **for** $s \in \mathcal{K}$ **do** $\Delta_s \leftarrow |C_s \setminus U|$; $\quad U \leftarrow U \cup C_s$
 3: **Update contributing set:** $\mathcal{A}_t \leftarrow \{s \in \mathcal{K}, \Delta_s > \tau\}$
 4: **Normalize coverage gains:** $\widehat{\Delta}_s \leftarrow \Delta_s \big/ \sum_{j \in \mathcal{A}_t} \Delta_j$
 5: **Construct coverage-weighted teacher:** $Q_t(x) \leftarrow \sum_{s \in \mathcal{A}_t} \widehat{\Delta}_s \, p_s(x)$
 6: **Form blended target for next epoch:** $\tilde{y}_{t+1}(x) \leftarrow (1 - \beta_t) \, \text{one-hot}(y) + \beta_t \, Q_t(x)$
 7: **Return** $\tilde{y}_{t+1}(x)$

---

## B  COMPATIBILITY WITH DATA AUGMENTATION

To verify compatibility with common augmentations, we tested *CAFÉ* using CutMix, a widely adopted augmentation for image classification, with a ResNet-18 backbone. *CAFÉ* maintains its accuracy gains under CutMix, confirming that its coverage-aware distillation integrates smoothly with representative, state-of-the-art augmentation strategies (see Table 4).

Table 4: CutMix. Mean test accuracy with and without CutMix augmentations on CIFAR100.

| Method | CIFAR-100 Symmetric | | | CIFAR-100 Asym. | | TinyImageNet |
| | Clean | 20% | 40% | 20% | 40% | Clean |
|---|---|---|---|---|---|---|
| Vanilla + Early Stopping | $78.66 \pm .11$ | $64.98 \pm .11$ | $59.37 \pm .36$ | $66.61 \pm .12$ | $49.57 \pm .10$ | $65.37 \pm .01$ |
| + CutMix | $\mathbf{80.75} \pm .16$ | $73.60 \pm .15$ | $67.15 \pm .46$ | $75.10 \pm .21$ | $62.81 \pm .31$ | $68.45 \pm .19$ |
| *CAFÉ* | $80.25 \pm .05$ | $74.09 \pm .01$ | $\mathbf{69.75} \pm .14$ | $75.61 \pm .18$ | $67.17 \pm .10$ | $68.91 \pm .01$ |
| + CutMix | $80.74 \pm .09$ | $\mathbf{74.45} \pm .19$ | $69.19 \pm .30$ | $\mathbf{75.95} \pm .12$ | $\mathbf{71.47} \pm .16$ | $\mathbf{70.62} \pm .04$ |

## C  ABLATION STUDY: FULL RESULTS

### C.1  *CAFÉ*-INSPIRED EMA

Classical Exponential Moving Average (EMA) smooths parameter updates by maintaining a running average of the model weights across epochs. In order to align this idea with our coverage notion, we devised a variant of EMA that takes advantage of the scores used by *CAFÉ*, replacing mere temporal averaging by reweighting past checkpoints according to their *marginal validation coverage*. Specifically, at the end of each epoch we compute the incremental validation coverage contributed by each earlier checkpoint, normalize these increments into weights, and construct a teacher model in parameter space by averaging the corresponding checkpoints. The current model is then blended towards this teacher, yielding the method termed *CAFÉ*-EMA.

Table 5 summarizes these results for CIFAR-100 (clean). Both Vanilla EMA and *CAFÉ*-EMA ($2^{nd}$ and $4^{th}$ rows) use the same Cross Entropy loss as in Vanilla ($1^{st}$ row); the only difference lies in the dynamic adjustment of network weights via EMA. Vanilla EMA (decay = 0.999) performs slightly worse than Vanilla ($78.32 \pm 0.14$ vs. $78.66 \pm 0.11$), and the *CAFÉ*-EMA variant underperforms further at $76.60 \pm 0.23$. In contrast, Basic *CAFÉ* achieves $80.25 \pm 0.05$. This suggests that while marginal coverage weighting is effective for soft label formation in self-distillation, applying it

Table 5: *CAFÉ*-inspired EMA ablation on CIFAR-100 (clean).

| Method | CIFAR-100 |
|---|---|
| Vanilla + Early Stopping (ES) | $78.66 \pm .11$ |
| Vanilla EMA + ES (decay = 0.999) | $78.32 \pm .14$ |
| *CAFÉ* | $\mathbf{80.25} \pm .05$ |
| *CAFÉ*-EMA | $76.60 \pm .23$ |

Table 6: Joint ablation results on CIFAR-100 with 40% label noise.

| Method | Sym 40% | Asym 40% |
|---|---|---|
| Vanilla + Early Stopping | $59.37 \pm .36$ | $49.57 \pm .10$ |
| Accuracy-only weighting (C.2) | $67.73 \pm .16$ | $62.37 \pm .34$ |
| Chronological coverage (C.3) | $\mathbf{69.49} \pm .31$ | $64.18 \pm .32$ |
| Temporal smoothing (C.4) | $69.37 \pm .18$ | $64.50 \pm 1.30$ |
| *CAFÉ* | $\mathbf{69.75} \pm .14$ | $\mathbf{67.17} \pm .10$ |

directly in parameter-space (i.e., averaging and blending the network weights themselves across checkpoints, rather than averaging predictions or soft labels) does not yield the same benefits, and may even hinder optimization. This confirms that validation-coverage weighting is most effective when applied in the *prediction space*, rather than in the parameter space.

### C.2 COVERAGE VS. ACCURACY-ONLY AVERAGING

We now consider a variant that *ignores coverage entirely* and uses the *normalized validation accuracy* of each checkpoint as a direct weight in the teacher, without considering marginal coverage. The key question is whether *CAFÉ*'s advantage arises from its *coverage-aware credit* or simply from *favoring high-accuracy checkpoints*. Results are reported in Table 6, showing that *CAFÉ* surpasses the accuracy-only variant under both types of noise, further indicating that **coverage-aware credit assignment is crucial**.

### C.3 CHRONOLOGICAL COVERAGE VS. ACCURACY-SORTED MARGINAL COVERAGE

We investigate an alternative choice for *CAFÉ* where we retain the coverage notion but replace Step 1 in Algorithm 1 with a simple chronological ordering: checkpoints are sorted by time so that later checkpoints come first when assigning credit. This contrasts with *CAFÉ*, which sorts checkpoints by validation accuracy and uses marginal coverage in that order. Results are reported in Table 6, showing that *CAFÉ* is superior or on par for both noise settings, validating our decision to base marginal coverage on accuracy-sorted checkpoints rather than chronological order.

### C.4 TEMPORAL SMOOTHING OF SOFT LABELS

The current ablation applies a smoothing window (of size 3) across adjacent checkpoints before forming the teacher, aiming to reduce noisy per-epoch fluctuations. The question is whether *CAFÉ*'s validation-guided ordering and marginal coverage already provide sufficient stability. Results are reported in Table 6, showing that *CAFÉ* remains superior across both noise types, suggesting that validation-guided marginal coverage already yields robust soft labels, with no extra benefit from local temporal smoothing.

### C.5 LIGHT-*CAFÉ* VS. BASIC *CAFÉ*

The light version of *CAFÉ*, introduced in Section 3.3, is designed to be substantially more efficient than the basic variant described in Section 3.1, as confirmed by the complexity analysis in Section 5. To assess the trade-off in practice, we compare the two methods empirically.

Table 7 shows that the accuracy of the two methods remains virtually identical when the tolerance parameter of Light *CAFÉ* is set to $\tau = 0.01$. Further evidence is provided in Figure 6, which tracks the size of the contributing set $\mathcal{A}_t$ (see Appendix A, Alg. 2) throughout training. The largest value observed along each curve essentially represents the amount of checkpoint storage required to execute the entire run end-to-end. With $\tau = 0.01$, these maxima drop from 173–176 checkpoints down to just 7–12 across the three different noise settings. Despite these drastic reductions, accuracy remains essentially unchanged (Table 7), confirming that *CAFÉ*'s advantage stems from the quality of the historical signals that are being retained, not from maintaining a large archive.

Table 7: **Accuracy and maximum contributing checkpoints for Basic and Light *CAFÉ*.** Tolerance of $\tau=0.01$ prunes low-coverage checkpoints, slashing storage needs while preserving accuracy.

| Method | Clean | | 40% Symmetric | | 40% Asymmetric | |
|---|---|---|---|---|---|---|
| | Acc. (%) | Required ckpt space (#) | Acc. (%) | Required ckpt space (#) | Acc. (%) | Required ckpt space (#) |
| CAFÉ | $80.25 \pm .05$ | 173 | $69.75 \pm .14$ | 176 | $67.17 \pm .10$ | 176 |
| *Light* CAFÉ ($\tau=0.01$) | $80.08 \pm .15$ | 7 | $69.55 \pm .15$ | 12 | $66.88 \pm .21$ | 12 |

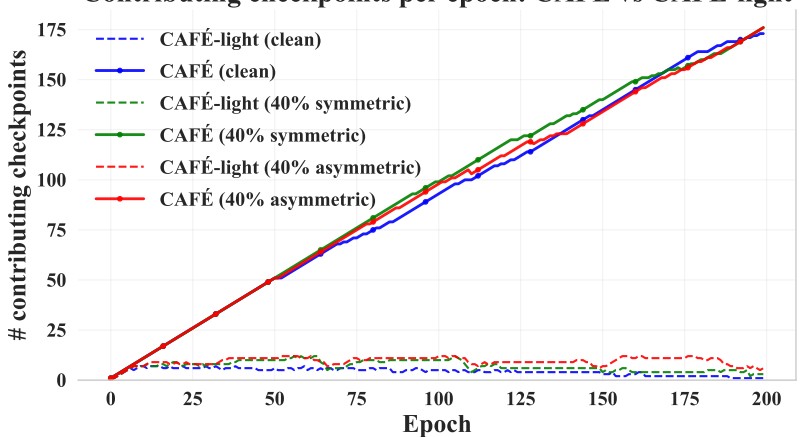

**Contributing checkpoints per epoch: CAFÉ vs CAFÉ-light**

Figure 6: Contributing checkpoints per epoch on CIFAR-100. Solid lines correspond to Basic *CAFÉ* and dashed lines to Light *CAFÉ* with tolerance threshold $\tau=0.01$. The peak value of each curve indicates the maximum checkpoint storage needed for a full run.

### C.6 EFFECT OF THE $\beta$ SCHEDULE

We now study the effect of the $\beta$ schedule used to interpolate between hard labels and the teacher distribution in *CAFÉ*. Recall that in the main method we define a normalized epoch index

$$u_t \;=\; \frac{t-1}{E-1} \in [0,1], \tag{4}$$

and use an exponential schedule of the form

$$\beta_t^{(k)} \;=\; \frac{1 - \exp(-k\,u_t)}{1 - \exp(-k)}, \tag{5}$$

where the default choice in *CAFÉ* is $k = 2$. In this ablation we vary $k$ in this exponential family (while keeping all other hyperparameters fixed), and also compare against simple baselines: constant schedules $\beta_t \equiv c$ with $c \in \{0, 0.5, 1\}$, a linear schedule $\beta_t = u_t$, and a cosine schedule $\beta_t = \frac{1-\cos(\pi u_t)}{2}$.

Table 8 reports test accuracy on CIFAR-100 with 40% symmetric noise. Constant schedules provide useful reference points: fix-0 corresponds to relying entirely on hard labels with no teacher signal, whereas fix-1 corresponds to relying solely on the teacher distribution. Both extremes perform substantially worse than any gradually increasing schedule. In contrast, smooth schedules such as linear, cosine, or exponential ($k = 1, 2, 3$) all yield similar accuracies, showing that the important design choice is to let $\beta_t$ increase over training, and that *CAFÉ* is robust to the exact functional form of this increase.

Table 8: Effect of the $\beta$ schedule on CIFAR-100 with 40% symmetric noise. We report mean test accuracy (%) and standard error. Smooth increasing schedules (linear, cosine, exponential with different $k$ values) behave similarly, while constant schedules perform substantially worse.

| Schedule | Test accuracy (%) |
|---|---|
| fix-0 | $59.39 \pm .41$ |
| fix-0.5 | $61.68 \pm .38$ |
| fix-1 | $0.94 \pm .06$ |
| linear | $69.26 \pm .08$ |
| cosine | $69.36 \pm .02$ |
| exp; $k = 1$ | $69.23 \pm .15$ |
| exp; $k = 2$ (default) | $69.75 \pm .14$ |
| exp; $k = 3$ | $69.53 \pm .59$ |

# D FULL *CAFÉ* ANALYSIS

## D.1 COMPLEXITY DERIVATION

To analyze the complexity of Alg. 1, let $n = |S_t|$ and $m$ the size of the validation set.

**Time complexity:**

i) Step 1 is dominated by $\mathcal{O}(n \log n)$.

ii) Step 2 is dominated by $\mathcal{O}(nm)$.

iii) Steps 3-4 are dominated by $\mathcal{O}(n)$.

*Overall time complexity* is $\mathcal{O}(n \log n + nm)$.

**Space complexity:** Storing predictions $\{p_s(x)\}$ for $n$ checkpoints requires $\mathcal{O}(n)$.

Similarly, the time and space complexity of Alg. 2 in the appendix is dominated by the size of the set of *contributing checkpoints* $\mathcal{A}_t \subseteq S_t$, where $|\mathcal{A}_t| \leq |S_t|$. The larger the tolerance parameter $\tau$ is, the larger the decrease in complexity achieved by Light *CAFÉ* as compared to Basic *CAFÉ*. Our empirical study shows that there is already a large gain even for $\tau = 0.01$.

## D.2 MARGINAL COVERAGE CLAIM: FORMAL STATEMENT AND PROOF

For any checkpoint $s$, if there exists an epoch $t$ at which its marginal coverage is null, i.e. $\Delta_s = 0$, then $\Delta_s = 0$ for all subsequent epochs. Consequently, the space complexity is bounded by the size of the set of contributing checkpoints for which $\Delta_s > 0 \; \forall t$ – new checkpoints may be added to this set, but once a checkpoint is discarded, it cannot reenter.

This is stated more formally in the next claim:

**Claim 1.** *Let $S_t = \{s_1, \ldots, s_t\}$ denote the ordered set of checkpoints up to epoch t, ranked by the size of their coverage subset $C_{s_i}$ - the set of points that checkpoint $s_i$ correctly classifies in the validation set. Using (1), define the marginal coverage of checkpoint $s_i$ at epoch t*

$$\Delta_{s_i}^t = \big| C_{s_i} \setminus \bigcup_{j=1}^{i-1} C_{s_j} \big|.$$

*Let $\mathcal{B}_t \subseteq S_t$ denote the support set of checkpoints at epoch t, defined as*

$$\mathcal{B}_t = \{ s_i \in S_t \mid \Delta_{s_i}^t > 0 \}.$$

*Given an epoch t and some future epoch $T > t$, $\mathcal{B}_T \cap \{S_t \setminus \mathcal{B}_t\} = \emptyset$.*

*Proof.* By contradiction. Suppose there is a checkpoint $s$ and epoch $T$ such that $s \in S_{T-1} \subset S_T$. Assume that $s \in \{S_{T-1} \setminus \mathcal{B}_{T-1}\} \implies s \notin \mathcal{B}_{T-1}$ and $s \in \mathcal{B}_T$. Consider 2 cases:

1. $|C_s| \geq |C_T|$: By construction checkpoint $T$ succeeds checkpoint $s$ in both ordered sets $S_{T-1}$ and $S_T$. Therefore the marginal coverage $\Delta_s$ in both epochs is the same, and $s \notin \mathcal{B}_{T-1} \implies s \notin \mathcal{B}_T$, in contradiction to $\mathcal{B}_{T-1} \cap \{S_{T-1} \setminus \mathcal{B}_{T-1}\} = \emptyset$.

2. $|C_s| < |C_T|$: Let $\tilde{S}$ denote the set of checkpoints that precede $s$ in ordered set $S_{T-1}$. It follows that the set of checkpoints that precede $s$ in ordered set $S_T$ is $\tilde{S} \cup \{T\}$.

$$s \notin \mathcal{B}_{T-1} \implies \Delta_s^{T-1} = 0 \implies C_s \subseteq \bigcup_{q \in \tilde{S}} C_q \subseteq \bigcup_{q \in \tilde{S} \cup \{T\}} C_q$$

$$s \in \mathcal{B}_T \implies \Delta_s^T \neq 0 \implies C_s \not\subseteq \bigcup_{q \in \tilde{S} \cup \{T\}} C_q$$

which is a contradiction.

$\square$

## E   VALIDATION ACCURACY AS A SURROGATE FOR GENERALIZATION

We next justify using validation accuracy as the weighting signal in *CAFÉ*. Figure 7 shows standard vanilla learning curves (without *CAFÉ*) across multiple noise distributions and learning-rate schedules. Although label noise can cause slight divergence between validation and test sets, their accuracies still track each other closely under all regimes. This consistent co-variation confirms that validation accuracy reliably reflects generalization trends, even when labels are corrupted. Consequently, marginal coverage weights derived from validation scores enable *CAFÉ* to recognize forgotten patterns and propagate useful knowledge forward without ever referencing the test set.

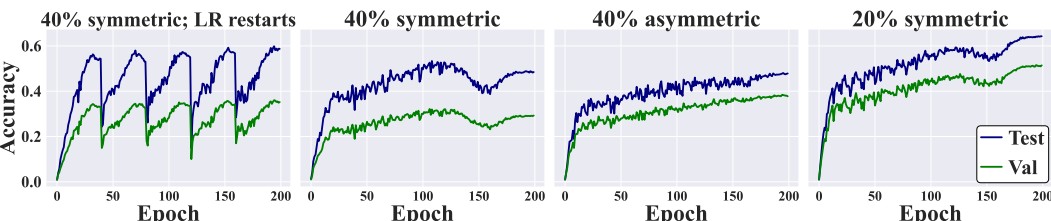

Figure 7: Test vs. validation accuracy over time for CIFAR-100 under 4 settings (from left to right): 1) 40% symmetric noise with learning-rate restarts; 2) 40% symmetric noise (monotonic LR); 3) 40% asymmetric noise; and 4) 20% symmetric noise. Blue curves show test accuracy, noting that the test data lacks label noise (thus the high accuracy values). The green curves show validation accuracy, noting that validation data has the same label noise distribution as the training data.

## F   IMPLEMENTATION DETAILS

We conducted experiments on three image classification datasets: CIFAR-100 (Krizhevsky et al., 2009), TinyImageNet (Le & Yang, 2015), and CIFAR-100N Wei et al. (2022).

On CIFAR-100 and TinyImageNet without label noise, all models were trained for 200 epochs with batch size 32, learning rate 0.01, SGD (momentum 0.9, weight decay 5e-4), cosine annealing, and standard augmentations (horizontal flip, random crop). For noisy label experiments, we used cosine annealing with warm restarts (every 40 epochs), a 0.1 learning rate updated per batch, and a batch size of 64.

To evaluate the impact of blending soft and hard labels, we conducted experiments using a variety of $\beta$ schedules - both constant and varying - to govern the final mixture,

$$\tilde{y}_{t+1}(x) = (1 - \beta_t)\,\text{one-hot}(y) + \beta_t\,Q_t(x).$$

To avoid tuning $\beta$ specifically for optimal performance, we first tested candidate functions on a small subset of the data and selected the most promising configuration, which was then applied to the full evaluation as reported in Section 4.

For a fair comparison, we re-trained the SAT method (Huang et al., 2020) shown in Tables 1 and 2 using the same architecture, data, and training scheme as our approach, following the additional guidelines in the paper. As this method is designed to integrate with existing training schemes, this setup ensures consistency. All experiments were run on A5000 GPUs.

## G  INJECTING LABEL NOISE

For label noise experiments, we injected noise using two standard methods (Patrini et al., 2017):

1. **Symmetric noise:** a fraction $p$ of labels is randomly selected and replaced uniformly with a different label.

2. **Asymmetric noise:** a fraction $p$ of labels is randomly selected and altered using a fixed label permutation.

## H  LLM USAGE

Large language models were used in a limited, assistive role for phrasing and discovery of related work; all surfaced references were manually verified. The models did not contribute novel ideas or technical content; all methods, analyses, experiments, and writing decisions are the authors' own, and the authors take full responsibility for the paper.

