# OpenReview forum: "CAFÉ: Coverage-Aware Self-Distillation to Mitigate Forgetting in Deep Networks"
_ICLR.cc/2026/Conference — Submitted to ICLR 2026_

### Official Review · Reviewer_A8Am · 2025-10-29

**Soundness:** 2
**Presentation:** 2
**Contribution:** 2
**Rating:** 4
**Confidence:** 4

**Summary:**

This paper introduces CAFÉ (Coverage-Aware Forgetting
Elimination), a method aimed at reducing local forgetting in deep network training. The approach constructs a teacher distribution by weighting past checkpoints according to their *marginal validation coverage*, thus emphasizing models that uniquely contribute to correctly predicted validation samples. This self-distillation mechanism helps preserve useful past knowledge without ensemble inference or additional model parameters. Experiments on CIFAR-100, CIFAR-100N, and TinyImageNet show consistent gains over strong baselines such as SAT and early stopping, and an efficient variant (Light CAFÉ) maintains similar performance with much lower storage cost.

**Strengths:**

1. The proposed CAFÉ framework is simple, effective, and model-agnostic, requiring no architectural changes or additional inference time cost.
2. The paper addresses an important problem (i.e., local forgetting during standard supervised training) with clear motivation and empirical evidence.

**Weaknesses:**

1. The method requires storing multiple checkpoints during training (even if Light CAFÉ mitigates this), which could be burdensome for large-scale models.
2. The experiments are limited to CNN architectures (ResNet, DenseNet). It remains unclear whether the observed local forgetting phenomenon also occurs in Transformer-based models such as ViT or Swin Transformers, which are now dominant in vision tasks. If such forgetting exists, would the proposed CAFÉ still be effective, or would it require adaptation to the Transformer training dynamics?
3. The analysis of “local forgetting” is primarily demonstrated on CIFAR-100 (and its noisy variant). It is unclear whether the same phenomenon appears on large-scale datasets such as ImageNet, where the data distribution is more diverse and the training is typically longer and more stable. Is local forgetting a general property of deep network training, or is it amplified by small datasets and limited data diversity? Moreover, would CAFÉ still provide benefits when the model already achieves high coverage on large datasets?
4. The paper lacks a direct comparison to other forgetting mitigation techniques in continual learning outside self-distillation (e.g., rehearsal-based or regularization-based methods).

**Questions:**

Please see the Weaknesses section above.

---

> ### Author Response · Authors · 2025-12-03
>
> We thank the reviewer for the thoughtful comments and for acknowledging the motivation, clarity of the framework, and the empirical strength of CAFÉ. We address the concerns below.
>
> ---
>
> **1. “The method requires storing multiple checkpoints during training.”**
>
> This is a reasonable observation. Although the conceptual description of CAFÉ refers to previous checkpoints, the actual implementation does not store checkpoint weights. Instead, it stores only the corresponding training predictions, which are already computed during training. This makes the method invariant to model size since prediction tensors scale only with dataset size, not with parameter count. These stored predictions are needed only during training and can be discarded afterward. Combined with the storage reduction achieved by the Light CAFÉ variant, which lowers memory usage by approximately ninety percent, the method remains practical even for large-scale models.
>
> ---
>
> **2–3. “Experiments are limited to CNN architectures (ResNet, DenseNet). It is unclear whether local forgetting appears in ViT or Swin…,” and “Local forgetting is demonstrated only on CIFAR-100… does it also occur on ImageNet?”**
>
> We appreciate these related questions and address them together for clarity.
>
> The concept of local overfitting is not introduced for the first time in our work; it is already established in prior work, including the Knowledge Fusion paper, which demonstrates the same phenomenon in modern architectures such as ViT. Since our goal is not to redefine the phenomenon but to provide a more practical and online alternative to a post-hoc ensemble, we build on this existing evidence.
>
> KF also shows that validation-driven coverage signals transfer well to ViTs, suggesting that this family of methods is architecture agnostic. Given that CAFÉ follows the same conceptual motivation while achieving consistent gains across CIFAR-100, CIFAR-100N, and TinyImageNet under highly varied conditions, we have strong reason to believe that the method would behave similarly on ViT and on larger datasets such as ImageNet. We agree that including such experiments could strengthen the final paper and acknowledge this as promising future work.
>
> ---
>
> **4. “The paper lacks comparison to continual learning methods.”**
>
> Continual learning operates under a fundamentally different problem formulation, where tasks arrive sequentially and the objective is to avoid catastrophic forgetting across tasks. Our work studies within-task forgetting that occurs during standard end-to-end supervised training with a fixed label space.
>
> Because the goals, assumptions, and evaluation settings differ substantially, direct comparison to task-sequential methods would not be meaningful. Our baselines therefore focus on the strongest methods designed for the same single-task supervised training paradigm, including SAT, PS-KD, and KF.

---

### Official Review · Reviewer_PEp2 · 2025-10-30

**Soundness:** 3
**Presentation:** 2
**Contribution:** 3
**Rating:** 4
**Confidence:** 2

**Summary:**

This paper proposes CAFÉ,  a self-distillation method that mitigates local forgetting in deep neural networks. CAFÉ weights previous training checkpoints based on their marginal validation coverage, a metric that measures how many validation examples a checkpoint gets right that others miss. With respect to other methods that solve the problem by using an ensemble of previous checkpoint, at the end Cafè is able to produce a single model. This coverage-weighted knowledge is distilled into the current model, forming adaptive soft targets that preserve useful information over time. Experiments are conducted on CIFAR-100 and TinyImageNet where the proposed method is compared against self-distillation and ensemble methods, under both clean and noisy labels.

**Strengths:**

- The idea of using previous checkpoints during training is interesting, as it allows the method, unlike other approaches that rely on checkpoint ensembles, to produce a single model at the end of training.
- The paper also proposes a lightweight version that reduces storage requirements by limiting the number of checkpoints, without compromising performance.
- As shown in Fig. 2, the method effectively mitigates the problem of local forgetting during training, whereas other methods address it only post hoc

**Weaknesses:**

- The results are somewhat incremental with respect to KF in Tables 1 and 2, and with respect to PS-KD in Table 3.
- Style of the paper: The paper lacks clarity in the presentation of results. Table 3 shows several rows missing. Figure 4 was obtained by overlaying results on top of an image taken from another paper (as clearly stated in lines 378–381). The last row of Tables 1 and 2 is misleading, as it initially appears to show improvement over the state of the art, while it actually compares against ERM + early stopping. Line 79 includes the phrase “see review below” without directly referencing the relevant section.
- Experiments using more recent vision backbones (e.g., ViT) are missing. see questions.
- Figure 5b is unclear and difficult to interpret. Please clarify the main message in the caption, even if it is already discussed in the main text.
- The title can be misleading as usually forgetting refers to forgetting of previous knowledge in the Continual Learning setting. I would clarify that.

**Questions:**

- What is the main difference between CAFE and FK? As far as I understand, FK also uses a similar strategy involving previous checkpoints. The main difference (and advantage) of CAFE is that the proposed method does not require an ensemble during inference.
- Since Table 3 shows that PS-KD achieves results similar to CAFE, it would be useful to highlight the main differences between the two methods.
- In the KF paper, experiments were also conducted using more recent visual backbones such as ViT. Why are these missing from the proposed work?

---

> ### Author Response · Authors · 2025-12-03
>
> We thank the reviewer for the careful reading of the paper and for the constructive feedback.
> For clarity, we separate our response into two parts: (1) weaknesses and (2) questions.
> Below we address each weakness in turn.
>
> ---
>
> **1. “The results are somewhat incremental with respect to KF in Tables 1 and 2, and with respect to PS-KD in Table 3.”**
>
> We appreciate the opportunity to clarify the significance of the gains.
> The comparison to PS-KD appears only in the clean CIFAR-100 setting, which is widely recognized as one of the most challenging regimes in which to obtain measurable improvements within the single-network training paradigm. Even under these conditions, our method improves performance across all three architectures evaluated. For instance, the ResNet-18 error decreases from 20.82 to 19.75, a substantial and statistically robust reduction of 1.05 points. Relative to effect sizes reported by prior self-distillation work, these gains are comparable or larger.
>
> Across noisy-label settings, the improvements over KF are even more pronounced.
> Averaged over all settings in Tables 1 and 2, our method improves over KF by **1.63 points** while requiring only a single model produced in one training run. KF, in contrast, depends on a large checkpoint ensemble and incurs substantial storage and inference cost. Reducing the required resources while simultaneously improving accuracy is a central motivation of our work and a key distinction.
>
> ---
>
> **2. “The paper lacks clarity in the presentation of results.”**
>
> We appreciate this suggestion and provide clarifications:
>
> &nbsp;&nbsp;&nbsp;&nbsp;(a) *Missing entries in Table 3.*
> Many prior methods reported results only for a subset of architectures or metrics.
> We opted to aggregate all available numbers from the literature rather than omit methods entirely, so that readers can compare all published baselines in one unified table.
>
> &nbsp;&nbsp;&nbsp;&nbsp;(b) *Figure 4 overlay.*
> The original paper presented results exclusively as a bar-style plot without providing numeric values.
> We reproduced their presentation and overlaid our results to enable direct comparison.
>
> &nbsp;&nbsp;&nbsp;&nbsp;(c) *Final row of Tables 1 and 2.*
> This row has been updated to explicitly state that the gains are computed over the Vanilla baseline, removing any possible ambiguity. The state-of-the-art rows remain directly above, preserving transparency.
>
> &nbsp;&nbsp;&nbsp;&nbsp;(d) *Missing reference at line 79.*
> This has been corrected in the revision.
>
> ---
>
> **3. “Experiments using more recent vision backbones (e.g., ViT) are missing.”**
>
> This is a reasonable point. Running full experiments on newer large-scale backbones required compute and time that were not available within the submission window.
> We emphasize, however, that the validation-guided coverage signal underlying our method has already been shown in prior work (KF) to transfer smoothly to ViT architectures. Since our method relies on the same principle, but applies it in an online and more efficient form, we are confident that the behavior will extend to recent backbones as well.
> The consistent improvements observed across CIFAR-100, CIFAR-100N, and TinyImageNet under a wide range of noise conditions further support the robustness of the approach. Evaluating newer architectures remains an interesting extension that can complement the current results.
>
> ---
>
> **4. “Figure 5b is unclear and difficult to interpret.”**
>
> We appreciate this comment and have revised the caption to clearly communicate the main takeaway.
>
> ---
>
> **5. “The title can be misleading as usually forgetting refers to catastrophic forgetting in continual learning.”**
>
> We thank the reviewer for this point.
> Our work studies forgetting **within a single supervised task**, not catastrophic forgetting across tasks. This distinction is stated in the abstract, introduction, formal definition of the forget gap, and related work. The notion of forgetting here concerns a subset of examples from the *same* task that become misclassified as training progresses.
>
> ---

---

> > ### Author Response · Authors · 2025-12-03
> >
> > We now address the reviewer’s specific questions.
> >
> > ---
> >
> > **1. “What is the main difference between CAFE and KF?”**
> >
> > KF aggregates predictions from many stored checkpoints and uses them as an ensemble.
> > This provides strong performance but requires substantial storage and multiple forward passes at inference.
> > In contrast, our method produces a single model in one training run, requires no checkpoint ensemble at inference, and weights checkpoints by marginal validation coverage rather than the forget gap. Our method also integrates this information online through self-distillation, which KF does not perform. The two approaches therefore differ in training mechanism, weighting criterion, and final delivered model.
> >
> > ---
> >
> > **2. “Since Table 3 shows that PS-KD achieves results similar to CAFE, it would be useful to highlight the main differences.”**
> >
> > We appreciate this suggestion.
> > PS-KD averages past predictions uniformly or using predefined decay rules, without incorporating validation-based coverage or forgetting signals.
> > Our method instead uses checkpoint-specific marginal coverage, which highlights checkpoints that recover forgotten examples. This leads to stronger performance across the broader range of datasets and noise conditions in Tables 1 and 2.
> >
> > The revised paper clarifies this distinction.
> >
> > ---
> >
> > **3. “Why are experiments on more recent backbones such as ViT missing?”**
> >
> > As noted above, these experiments could not be completed within the submission timeline.
> > We agree that extending the evaluation to recent backbones may further strengthen the study, and we view this as a worthwhile future direction that could provide additional validation of the method’s robustness.
> >
> > ---

---

### Official Review · Reviewer_FxrM · 2025-10-31

**Soundness:** 3
**Presentation:** 3
**Contribution:** 3
**Rating:** 6
**Confidence:** 3

**Summary:**

This paper considers the problem of local overfitting in deep learning, where previously learned patterns are forgot within training. The proposed method keeps all previous checkpoints and aggregates their predictions to build pseudo labels. Experimental results show the effectiveness of the proposed methods on small image datasets.

**Strengths:**

+ Addressing local overfitting in deep learning is an interesting research topic.

+ The proposed method is easily understandable.

+ The performance gain is consistent throughout experiments.

**Weaknesses:**

- No ablation study on the design choices, including hyperparameter tuning. For example, is the schedule for beta in L254 optimal?

- As the proposed method requires to keep all previous checkpoints and runs them to get their outputs, an analysis on the computational cost compared with baseline methods is required.

- In Figure 5, why the marginal coverage of Vanilla peaks in the middle? Following the idea in STEP 2: MARGINAL COVERAGE SWEEP, the best performing model should be chosen at first, which usually appear around the end of training. If not, then it implies that the learning rate schedule is simply suboptimal, and could be better by hyperparameter tuning. In other words, the comparison might not be fair, as the optimization of Vanilla appears to be not properly done.

- Citation format is problematic in some places, e.g., L329.

- Accuracy is not sufficient to catch the degree of forgetting.

**Questions:**

Please address concerns in Weaknesses above.

---

> ### Author Response · Authors · 2025-12-03
>
> We thank the reviewer for the constructive and detailed feedback. Below we address each weakness and question.
>
> ---
>
> **1. Ablation study on design choices and hyperparameters (including the beta schedule).**
> The paper already includes several ablations on the core design components of CAFÉ. Specifically, we compare against an EMA-based variant, evaluate the use of coverage versus accuracy as checkpoint weights, test chronological versus accuracy-ordered traversal of checkpoints, and analyze the temporal smoothing version of CAFÉ. These ablations directly target the fundamental structural decisions that define the method.
>
> Regarding the hyperparameter choice raised in the review, namely the beta schedule in L254, we agree that this is an important design component. We have prepared a comprehensive ablation evaluating multiple alternative schedules (constant schedules, linear, cosine, and different exponential ramps) to assess the sensitivity of CAFÉ to this choice. The results show that while fixed schedules that eliminate or fully replace the teacher perform poorly, all smooth increasing schedules yield similar accuracy. This confirms that the method is robust to the specific form of the beta schedule. The full ablation is included in the revised version.
>
> ---
>
> **2. Computational cost and efficiency relative to baselines.**
> Although the conceptual description highlights multiple checkpoint evaluations, the practical implementation of CAFÉ is considerably lighter. In practice, each checkpoint only needs to be evaluated once on the validation set at the end of its epoch. This is already performed by the baselines, such as Vanilla + Early Stopping, which evaluate validation accuracy at every epoch. For the training set, model outputs are simply stored as training progresses, so no extra forward passes are needed. Appendix D already provides a detailed space complexity analysis showing that the memory and time overhead remain modest and scale linearly with the number of checkpoints. As shown in the paper, this overhead is significantly reduced when using the Light-CAFÉ variant, which is explicitly designed to minimize storage and computation while preserving performance.
>
> ---
>
> **3. Interpretation of Figure 5 and the peak of Vanilla’s marginal coverage.**
> It is expected in the noisy label regime that early or mid-training checkpoints may achieve the highest validation performance, because later checkpoints often begin to memorize noisy labels. This effect is well documented in noisy-label learning and directly explains why the baseline we compare against, namely Vanilla with early stopping, also relies on a held-out validation split: the mid-training checkpoint that attains this peak is precisely the one selected by early stopping for evaluation. Our baseline protocol is identical across all noise levels, including clean data, and follows the standard training practices described in the implementation details. Adjusting optimization hyperparameters specifically for a given noise level or dataset would tailor the baseline in a way that compromises fairness and could lead to poor generalization. The observed mid-training peak therefore reflects the natural progression of noisy-label optimization rather than suboptimal tuning.
>
>
> ---
>
> **4. Citation formatting.**
> We thank the reviewer for pointing this out. The formatting in L329 and related locations has been corrected in the revised submission.
>
> ---
>
> **5. “Accuracy is not sufficient to catch the degree of forgetting.”**
> The paper explicitly includes a forgetting metric, namely the forget gap defined on line 64. This metric measures the fraction of samples that were correctly classified at epoch \(e\) but misclassified by the final model. In Section 4.4 (Forget Analysis), we use this metric to compare the degree of forgetting between Vanilla + Early Stopping and CAFÉ. The metric clearly shows that CAFÉ substantially reduces the forget gap relative to the baseline, complementing accuracy-based evaluations.
>
> ---
>
> We appreciate the reviewer’s thoughtful comments. The points raised have helped strengthen the clarity and completeness of the submission, and we will integrate the corresponding improvements into the final revised version.

---

### Official Review · Reviewer_kdK9 · 2025-11-03

**Soundness:** 2
**Presentation:** 2
**Contribution:** 2
**Rating:** 4
**Confidence:** 4

**Summary:**

This paper introduces CAFÉ (Coverage-Aware Forgetting Elimination), an online, validation-aware self-distillation method to mitigate local overfitting and forgetting in deep neural networks. CAFÉ dynamically identifies checkpoints with unique validation coverage and forms soft teacher targets for subsequent training epochs based on their marginal contributions. Extensive experiments demonstrate CAFÉ’s robustness to clean and noisy labels, outperforming standard ERM, prior self-distillation baselines, and specialized ensembles (such as Knowledge Fusion) on CIFAR-100, TinyImageNet, and CIFAR-100N. The work also includes in-depth ablations and theoretical analysis of its claims and complexity.

**Strengths:**

1. Interesting approach to handle forgetting with self distillation from previous checkpoints based on validation accuracy.
2. Relatively easy to understand approach.
3. Good performance improvement on the CIFAR datasets, especially in high noise cases.
4. Significant improvements shown in Table 3 for clean data.

**Weaknesses:**

1. Reliance on a validation set that is representative of the test set. In cases where the test set may be significantly different, this approach will not be effective.
2. Very low improvement on TinyImageNet for low (symmetric) or zero noise cases. (Table 2).
3. Very few compared methods in Table 1 and 2. Reduces the confidence in the overall effectiveness of the approach.
4. Could not find any forgetting measure or metric.
5. Since the objective is dealing with the "forgetting of previously learned patterns.", wont incremental learning experiments be a better judge of how good the approach is in dealing with forgetting.

**Questions:**

1. Discuss the reliance on a validation set when it is not fully similar to the test set
2. Why is the improvement very low on TinyImageNet for low (symmetric) or zero noise cases. (Table 2).
3. Are there no more recent papers that can be compared with in Table 1 and 2?
4. Could not find any forgetting measure or metric.
5. Wont incremental learning experiments be a better judge of how good the approach is in dealing with forgetting?

---

> ### Author Response · Authors · 2025-12-03
>
> We thank the reviewer for the constructive feedback and for noting both the clarity of the method and the performance gains across clean and noisy settings. We address each point below.
>
> ---
>
> **1. Reliance on a validation set.**
> In the noisy‐label regime, relying on a small held-out validation split is essential, since training purely on noisy labels without any reference often leads to severe overfitting and memorization of label noise. The baselines we compare to also depend on this validation split. For example, **Vanilla + Early Stopping** uses the same validation set to select the best checkpoint, and **SAT** uses a validation split for hyperparameter selection. CAFÉ therefore does not assume anything beyond what these baselines already require.
>
> In our method, the validation set is used only to measure *relative coverage* of checkpoints, while the same validation information is used by competing methods to guide model selection. **Appendix E** provides supporting evidence across multiple noise settings and corruption types and shows that validation and test accuracy track each other closely, which supports the reliability of using the validation set in this context.
>
> ---
>
> **2. Improvement on TinyImageNet in low or no noise.**
> The improvements in the clean and low-noise TinyImageNet settings are substantial. In the clean case, CAFÉ improves accuracy by **3.54 points**, corresponding to a **5.4\%** relative gain. Under symmetric noise of 20%, 40%, and 60%, the improvements are **6.88**, **8.58**, and **7.10** points, corresponding to **12.2\%**, **17.3\%**, and **17.7\%** relative gain. These gains are comparable to the improvements on CIFAR-100, where the average across settings is **8.35** points and **15.7\%** relative gain.
> The TinyImageNet results therefore reflect improvements of the same magnitude and cannot be characterized as “very low.”
>
> ---
>
> **3. Few comparison methods.**
> Not many methods directly address the noisy-label regime in the single-network supervised training paradigm we study. Among those that do, **Self Adaptive Training (SAT)** is, to the best of our knowledge, the best performing method and is regarded as the state of the art on CIFAR-100, CIFAR-100N, and TinyImageNet under the symmetric and asymmetric noise settings considered in our work. CAFÉ is designed within the same paradigm and consistently outperforms SAT across all noise levels and datasets in Tables 1 and 2. We also compare against **Knowledge Fusion ensembles** and show that CAFÉ achieves higher accuracy while being more efficient.
>
> ---
>
> **4. Forgetting measure.**
> The forgetting measure used in our work is defined on line 64, where we introduce the **forget gap**. This metric measures the fraction of test samples that were correctly classified at epoch \(e\) but misclassified by the final model. We explicitly use this measure in **Section 4.4 (Forget Analysis)** to compare the average forget gap of *Vanilla + Early Stopping* with that of CAFÉ, demonstrating that CAFÉ significantly reduces forgetting relative to the baseline.
>
> ---
>
> **5. Incremental learning as an evaluation protocol.**
> Continual or incremental learning deals with catastrophic forgetting across different tasks that arrive sequentially. In contrast, our work studies forgetting *within a single supervised task* with a fixed label space. The forget gap we analyze reflects loss of accuracy on the same task over the course of standard training, which is a different phenomenon from task interference in continual learning.
> In our view, continual-learning benchmarks are therefore not the appropriate evaluation setting for this work. Our goal is to improve supervised training under noisy labels and to reduce the within-task forget gap. Extending CAFÉ to continual-learning scenarios is an interesting possible direction for future work but is outside the scope of this submission.

---

### Meta-Review · Area_Chair_bv78 · 2026-01-06

**Summary:**

The paper proposes an interesting idea , but the evidence is not strong enough to show that it is:

a general phenomenon,

a practically useful solution, or

a clear improvement over existing forgetting-mitigation methods.

The work is weakened by limited experiments, unclear evaluation of forgetting, missing comparisons, and presentation issues, making it hard to judge the real impact of the contribution. So, I think it is not ready for publish at this stage.

**Reviewer Concerns:**

The improvements over prior work  are incremental rather than substantial, and the contribution does not clearly advance the state of the art.

Several tables and figures are unclear or potentially misleading

Experiments are limited to specefic architectures  and small datasets .

**Reviewer Scores:**

The overall score is below the acceptance threshold, and the rebuttal does not convincingly address the major concerns, particularly the limited novelty and the incomplete experimental validation.

---

### Decision · Program_Chairs · 2026-01-26

Reject